# Transient Detection of Rotor Asymmetries in Squirrel-Cage Induction Motors Using a Model-Based Tacholess Order Tracking

**DOI:** 10.3390/s22093371

**Published:** 2022-04-28

**Authors:** Erik Etien, Thierry Doget, Laurent Rambault, Sebastien Cauet, Anas Sakout, Sandrine Moreau

**Affiliations:** 1LIAS Laboratory, University of Poitiers, 86073 Poitiers, France; thierry.doget@univ-poitiers.fr (T.D.); laurent.rambault@univ-poitiers.fr (L.R.); sebastien.cauet@univ-poitiers.fr (S.C.); sandrine.moreau@univ-poitiers.fr (S.M.); 2LaSIE Laboratory UMR CNRS 7356, University of La Rochelle, 17042 La Rochelle, France; asakout@univ-lr.fr

**Keywords:** soft sensor, speed estimation, adaptive observer, order tracking, squirrel-cage induction motor

## Abstract

In this article, we propose to determine the dynamic model of a squirrel-cage induction motor from a reduced amount of information. An adaptive observer is also built from this model in order to obtain a speed estimation and to perform rotor fault monitoring by Tacholess Order Tracking (TOT). We also propose a generalization of the notion of angular sampling in order to adapt to this type of defect. The procedure is validated in the laboratory on a test bench dedicated to the study of rotor bar defects.

## 1. Introduction

The detection of faults in squirrel-cage induction motors from electrical measurements is currently a well-understood subject at constant speed. Frequency methods are very effective because the signatures of the main defects are well known. At variable speed and for data sampled as a function of time, classical Fourier analysis is impossible because the frequencies of the spectral components sought vary with this speed. Many surveys have been published on the diagnosis of electrical machines, including the case of variable speed [1,2,3,4,5]. In these review articles, the methods cited are essentially time/frequency methods. There are quite a few articles citing Order Tracking (OT) as a method in its own right [2,3,6,7,8]. The main reason is that order tracking techniques require knowledge of the motor mechanical position. Their application from electrical measurements alone is one of the techniques called Tacholess Order Tracking. The article [9] is the reference survey on TOT. It summarizes the various measurements, other than mechanical (vibrations, electrical currents, video), making it possible to perform angular sampling and order analysis. In [9], the section relating to electrical measurements from induction motors is relatively brief because the estimation of rotational speed from electrical measurements is considered a well-known problem. Indeed, this subject was widely discussed in the context of sensorless motor control in the 1980s and 1990s [10,11,12,13,14]. Two possible approaches can be distinguished. The signal approach looks for a spectral component linked to the rotational speed in the stator current. They generally exploit specific construction features (number of slots, for example), with or without signal injection [15]. The model approach uses currents, voltages and the motor model to define an adaptive observer [16,17] or a reference model structure (MRAS) [18,19]. The model approach provides a robust speed estimation and also allows us to estimate other quantities (flux, resistances, rotor time constants, etc.) [20]. However, it requires knowledge of the dynamic motor model, which, in practice, requires identification of the parameters [21,22]. In an industrial context, it is generally impossible to carry out specific tests. Furthermore, available information is often reduced (nameplate and sometimes manufacturer’s documentation if the motor is recent), which strongly compromises the possibility of establishing a dynamic model.

In this article, we propose to circumvent this difficulty by going through the motor steady-state model (also called static model) in order to deduce the dynamic model. Determining the steady-state model of a squirrel-cage induction motor from a reduced amount of information is a research topic that has given rise to a number of publications [23,24,25,26,27,28]. The objective is to avoid having to carry out the conventional tests (no-load test and locked-rotor test) necessary to determine these parameters. This work provides us with a number of methods, each with its advantages and disadvantages. In [27], the authors propose a comparative study and establish a combined method exploiting the best of existing methods. This static model can be used to estimate a dynamic model and then build an observer to estimate the motor speed in real time. This speed can be used to estimate the mechanical position and perform the analyzed signal angular re-sampling. The proposed procedure makes it possible to set up monitoring by order tracking on a squirrel-cage induction motor for which we have very little information. It is particularly suitable for deployment in an industrial environment.

In Section 2, we recall the methods for obtaining the motor steady-state model from a reduced amount of information. The link between the static and dynamic models is established and an experimental study determines the estimation method to be used to satisfy variable-speed operation. In Section 3, the estimated speed is used for an angular re-sampling of the current modulation terms. A specific angle is defined for the monitoring of rotor bar breakage type faults. The entire procedure is experimentally tested in Section 4.

## 2. Observer Design from a Reduced Amount of Information

### 2.1. Model Estimation in Steady State

We recall in Figure 1 the squirrel-cage induction motor classical static model.

This model is a per-phase equivalent model under the assumption that the three input voltages are balanced. The meanings of the different parameters are as follows:$\omega_{e}=2\pi f_{e}$: supply electrical angular frequency;$R_{1}$: stator winding resistance;$L_{1}$: leakage inductance of a stator winding;$R_{c}$: resistance modeling iron losses;$L_{m}$: magnetizing inductance;$R_{2}$: resistance of an equivalent rotor winding transferred to the stator;$L_{2}$: leakage inductance of a rotor winding transferred to the stator;$s=\frac{f_{sync}-f_{r}}{f_{sync}}$: synchronism frequency with $f_{sync}=\frac{f_{e}}{p}$ and *p* the number of pole pairs.

The information needed to implement the methods mentioned above is as follows:The motor nameplate (NP), which provides the mechanical power $P_{n}$, the line voltage $V_{n}$, the current $I_{n}$, the frequency $F_{e}$, the speed $N_{n}$, the power factor ${FP}_{n}$, the torque $T_{n}$ and the nominal efficiency $\eta_{n}$ of the machine;Additional manufacturer data such as the maximum torque $T_{m}$, the starting torque $T_{d}$ and the starting current $I_{d}$, the motor code and class in the NEMA standard (National Electrical Manufacturers Association), and additional operating points for power factor and efficiency (100%, 50%, and 25% of load).

Table 1 summarizes the data necessary for the most common methods for determining model parameters.

A comparison of these different methods was made in [27] and indicated that most of the methods had at least one discrepancy. The Lee and Guimarães methods presented many problems in the calculation of reactance, presenting either negative values or absurdly high values. On the precision, the analytical $R_{1}$ estimation from the Natarajan-Misra, Haque, and Guimarães methods obtained good performance. For elements $R_{2}$, $X_{1}$, and $X_{2}$, the other methods presented better results, except for $R_{2}$ estimation in the Haque and Lee methods. Small deviations in $R_{c}$ values resulting from the Natarajan-Misra and Haque methods, although larger deviations of constant losses ($P_{const}$), are observed, which means that an accurate $R_{c}$ estimate does not necessarily imply a good estimate of the constant losses. It is found that Lee’s method shows a failure to provide Xm results. Despite this, the Lee method has the best performance in calculating constant losses, as well as the Natarajan and the Guimarães methods. By combining the strengths of each method in terms of robustness and precision, a new method has been proposed, the “COMBINED” method [27]. Improvements were observed in the accuracy of the parameter values as well as in the robustness of the new method, since it showed no malfunctions on different motors. The COMBINED method proposes and verifies experimentally a combination of the different methods:$R_{1}$ estimation with the Guimarães method;$R_{2}$ estimation with the Natarajan-Misra method;$R_{c}$, $X_{m}$ estimation with Haque’s method;$X_{1}$, $X_{2}$ estimation with Amaral’s method;constant losses $P_{const}$ estimation with the Lee method.

In a steady-state context, it is therefore possible to decide on the use of the COMBINED method to obtain the engine model. In this article, we are interested in the dynamic model of the machine. It is in this context that we will compare the different methods available in the following paragraphs.

### 2.2. From Steady State to Dynamic Model

Squirrel-cage induction motor dynamic models have mainly been developed in the context of flux vector control. They differ mainly in the choice of state variables used (stator flux, rotor flux, stator currents) and the reference frame chosen (fixed with respect to the stator, linked to the rotor or stator fields, linked to the rotor). This results in a large number of possible models whose electrical quantities are generally represented in the form of a two-phase system equivalent to the three-phase system. The model chosen here exploits the state variables rotor flux and stator current. The dynamic model is developed in any reference frame rotating at the angular frequency $\omega_{g}$ in complex notation,
(1)$${\underline{u}}_{s}=R_{s}{\underline{i}}_{s}+\frac{d{\underline{\phi }}_{s}}{dt}+j\omega_{g}{\underline{\phi }}_{s}$$
(2)$$R_{r}{\underline{i}}_{r}+\frac{d{\underline{\phi }}_{r}}{dt}+j\left(\omega_{g}-\omega \right){\underline{\phi }}_{r}=0.$$
(3)$${\underline{\phi }}_{s}=L_{s}{\underline{i}}_{s}+L{\underline{i}}_{r}$$
(4)$${\underline{\phi }}_{r}=L_{r}{\underline{i}}_{r}+L{\underline{i}}_{s}$$
with
${\underline{u}}_{s}$, ${\underline{i}}_{s}$, ${\underline{\phi }}_{s}$: stator voltages, currents and fluxes;${\underline{i}}_{r}$, ${\underline{\phi }}_{r}$: rotor currents and fluxes;$R_{s}$: stator resistance;$R_{r}$: rotor resistance;$L_{s}$: cyclic stator inductance;$L_{r}$: rotor cyclic inductance;$L=\frac{3}{2}M_{sr}$: $M_{sr}$, maximum mutual inductance between a stator phase and a rotor phase;ω: mechanical angular frequency;$\omega_{g}$: angular frequency of the frame in which the equations are expressed.

The angular frequency $\omega_{g}$ can be chosen in several ways.
$\omega_{g}=\omega_{s}$: the frame is fixed on the field rotating at the synchronism angular frequency. This frame is commonly called Park’s frame and is used in vector control. In this frame of reference, the electric and magnetic quantities are continuous.$\omega_{g}=0$: the frame is set on stator phase {a}. This frame is commonly called the Concordia frame and is often used in observer synthesis. In this frame, the electrical and magnetic quantities are sinusoidal.

In order to determine the existing relationships between the steady-state model ($R_{1},R_{2}$, $L_{1}$, $L_{2}$ and $L_{m}$) and the dynamic model ($R_{s},R_{r}$, $L_{s}$, $L_{r}$ and *L*), we choose the Park frame. It is a frame rotating at the power supply frequency. In steady state, the representation of these vectors can be likened to the Fresnel representation compatible with the steady-state model of the machine. In the Park reference, Equations (1) and (2) are written: (5)$${\underline{u}}_{s}=R_{s}{\underline{i}}_{s}+\frac{d{\underline{\phi }}_{s}}{dt}+j\omega_{s}{\underline{\phi }}_{s}$$
(6)$$R_{r}{\underline{i}}_{r}+\frac{d{\underline{\phi }}_{r}}{dt}+j\left(\omega_{s}-\omega \right){\underline{\phi }}_{r}=0.$$

We write Equations (5) and (6) in steady state by canceling the derived terms.
(7)$${\underline{u}}_{s}=R_{s}{\underline{i}}_{s}+j\omega_{s}{\underline{\phi }}_{s}$$
(8)$$R_{r}{\underline{i}}_{r}+j\left(\omega_{s}-\omega \right){\underline{\phi }}_{r}=0.$$

For the stator, by combining (7) and (3), we obtain
(9)$${\underline{u}}_{s}=R_{s}{\underline{i}}_{s}+jL_{1}\omega_{s}{\underline{i}}_{s}+jL\omega_{s}\left({\underline{i}}_{r}+{\underline{i}}_{s}\right)$$
with $L_{1}=L_{s}-L$.

For the rotor, we divide both sides of Equation (8) by the slip $s=\frac{\left(\omega_{s}-\omega \right)}{\omega_{s}}$, and we obtain:(10)$$\frac{R_{r}}{s}{\underline{i}}_{r}+j\omega_{s}{\underline{\phi }}_{r}=0$$

By combining (10) and (4), we have
(11)$$\frac{R_{r}}{s}{\underline{i}}_{r}+jL_{2}\omega_{s}{\underline{i}}_{r}+jL\omega_{s}\left({\underline{i}}_{s}+{\underline{i}}_{r}\right)=0$$
with $L_{2}=L_{r}-L$.

Equations (9) and (11) correspond to a diagram equivalent to that of Figure 1 with the changes in variable: $R_{s}=R_{1}$, $R_{r}=R_{2}$, $L=L_{m}$, $L_{s}=L_{1}+L_{m}$ and $L_{r}=L_{2}+L_{m}$. We can therefore directly define the dynamic model of an induction machine from its static model, from a reduced amount of information. In the following, we implement an adaptive observer built on the dynamic model above. The objective is to estimate the speed of the machine in order to deduce the mechanical angle and then to carry out the angular sampling of the analyzed signal.

### 2.3. Adaptive Observer

Observers for squirrel-cage induction motors were developed in the 1980s/90s with the aim of achieving controls without mechanical sensors. There is a large number of possible observers depending on the state variables that are estimated: stator currents, stator fluxes, rotor fluxes. In the case of speed, which is also a state variable, its direct estimation is impossible because the mechanical equation is, by definition, unknown, particularly the resistive torque applied to the shaft. We therefore use an adaptation mechanism, making it possible to estimate the speed at the same time as the other state variables. In the following, we use the Kubota observer [16], which is now a classic observer. This observer is built around Equations (Equation 1)–(Equation 4), expressed in the fixed frame relative to the stator (Concordia frame), i.e., $\omega_{g}=0$. The estimated state variables are the stator current and the rotor flux. The model, expressed as a state representation, is given by:(12)$$\begin{array}{c} \frac{d}{dt}\left(\begin{array}{c} {\hat{\underline{i}}}_{s}\\ {\hat{\underline{\phi }}}_{r} \end{array}\right)=\left(\begin{array}{cc} A_{11} & {\hat{A}}_{12}\\ A_{21} & {\hat{A}}_{22} \end{array}\right)\left(\begin{array}{c} {\hat{\underline{i}}}_{s}\\ {\hat{\underline{\phi }}}_{r} \end{array}\right)+\left(\begin{array}{c} B1\\ 0 \end{array}\right){\underline{v}}_{s}+G{\underline{e}}_{i}\\ \hat{w_{r}}=K_{p}\epsilon +K_{i}\int \epsilon dt \end{array}$$
with: ${\hat{\underline{i}}}_{s}={\left[\begin{array}{cc} {\hat{i}}_{s\alpha } & {\hat{i}}_{s\beta } \end{array}\right]}^{T}$, ${\hat{\underline{\phi }}}_{r}={\left[\begin{array}{cc} {\hat{\phi }}_{r\alpha } & {\hat{\phi }}_{r\beta } \end{array}\right]}^{T}$, ${\underline{e}}_{i}={\left[\begin{array}{cc} e_{i\alpha } & e_{i\beta } \end{array}\right]}^{T}=\left[\begin{array}{cc} \left({\hat{i}}_{s\alpha }-i_{s\alpha }\right) & \left({\hat{i}}_{s\beta }-i_{s\beta }\right) \end{array}\right]$, and $\epsilon =e_{i\alpha }{\hat{\phi }}_{r\beta }-e_{i\beta }{\hat{\phi }}_{r\alpha }$.

The notation $\hat{\left(.\right)}$ symbolizes the estimated quantities. The matrices are defined by
$A_{11}=-\left(R_{s}/\left(\sigma L_{s}\right)+\left(1-\sigma \right)/\left(\sigma \tau_{r}\right)\right)I=a_{r11}I$${\hat{A}}_{12}=-L_{m}/\left(\sigma L_{s}L_{r}\right)\left(\left(1/\tau_{r}\right)I-{\hat{\omega }}_{r}J\right)=a_{r12}I+{\hat{a}}_{i12}J$$A_{21}=\left(L_{m}/\tau_{r}\right)I=a_{r21}I$${\hat{A}}_{22}=-\left(1/\tau_{r}\right)I+{\hat{\omega }}_{r}J=a_{r22}I+{\hat{a}}_{i22}J$$B_{1}=1/\left(\sigma L_{s}\right)I=b_{1}I$$G={\left(\begin{array}{cccc} g_{1} & g_{2} & g_{3} & g_{4}\\ -g_{2} & g_{1} & -g_{4} & g_{3} \end{array}\right)}^{T}$$I=\left(\begin{array}{cc} 1 & 0\\ 0 & 1 \end{array}\right)$, $J=\left(\begin{array}{cc} 0 & -1\\ 1 & 0 \end{array}\right)$

where
$\sigma =1-L_{m}^{2}/\left(L_{s}L_{r}\right)$: leakage coefficient;$\tau_{r}=L_{r}/Rr$: rotor time constant;${\hat{\omega }}_{r}$: mechanical motor angular frequency.

Observer tuning parameters are *G*, $K_{p}$, and $K_{i}$. In the rest of this article, we will take G=0, which means that the dynamics of the observer are equal to motor ones. The rotation speed estimation dynamics will be determined by the parameters $K_{p}$ and $K_{i}$.

### 2.4. Experimental Model Selection

We test the proposed methodology on a laboratory test bench (Figure 2). This bench was made to study rotor bar faults on squirrel-cage induction motors [29]. Figure 3 shows the heathy and faulty rotors. The motor characteristics are: power 1.1 kW, rated voltage 400/230 V, rated current 2.6/4.3 A, cos(phi) 0.85/0.82, rated speed 1425 rpm. The motor is coupled to a DC generator delivering on a resistive load. Placed at the end of the shaft, an incremental encoder at 1024 pulses/revolution provides a mechanical position measurement. This measurement is connected to the simulink/dSpace processing block, which calculates the velocity by derivative. This angular speed is filtered by a low-pass filter of order 1 in order to limit the noise. All the electrical and mechanical measurements are processed by an anti-aliasing filter before being acquired by a Matlab/dSpace 1104 system with a sampling frequency $F_{s}=10$ kHz.

From the motor nameplate and the motor manufacturer’s documentation, the various methods for determining the model in steady state are applied. Results are given in Table 2.

It is difficult to establish the best choice among the five methods proposed in Table 2 for the simple reason that, for this motor, we do not have any references from the manufacturer. The motor is old and of low power; the method of Guimarães seems to give a false estimate of $X_{2}$. This is also due to the fact that this method gives better results for motors with higher powers, beyond 7.5 kW ($R_{1}$ and $X_{1}$ values are then lower). For all the other methods, the estimated parameters remain of the same order of magnitude. The combined model is established according to the procedure defined in [27]. From these parameters, five dynamic models are defined and used to build five adaptive observers. In the following test, we study the behavior of the various observers at variable speed. The motor is started directly on the network. The generator is coupled to a resistor set so that, in steady state, the motor absorbs its rated current (Figure 4). The observer tuning parameters are G=0, $K_{p}=0$, $k_{i}=$ 10,000.

In order to compare dynamic models, we plot the instantaneous amplitudes and frequencies IA(t) and IF(t), calculated by Concordia (measurements) and estimated for each model:(13)$$IA\left(t\right)=\sqrt{i_{s\alpha }{\left(t\right)}^{2}+i_{s\beta }{\left(t\right)}^{2}}$$
(14)$$IF\left(t\right)=atan\left[\frac{i_{s\beta }\left(t\right)}{i_{s\alpha }\left(t\right)}\right]$$

The result is given in Figure 5.

First, we note that in steady state and for this motor, the COMBINED and Haque methods give the better estimations. On the other hand, in transitory mode, one notices that the methods of Lee and Haque give the most precise estimate. The COMBINED method is not, on this test, the most precise in the estimation of IA(t) and IF(t). In the context of the TOT, we are particularly interested in the estimation of the speed. Figure 6 shows the estimates obtained for the different models. The error is calculated by simply taking the difference between the measured speed and the estimated speed.

It is verified that in steady state (top left of the figure), the COMBINED method gives the best result, which is verified by plotting the error in steady state (bottom right). On the other hand, in the transient regime, the COMBINED method is not the most precise. We find again, as in the case of IA(t) and IF(t) estimates, that the methods of Lee and Haque provide the best estimate. These results can be verified by calculating the rms value of the various errors in the transient and steady-state regions:(15)$$x\_rms\_err=\sqrt{\frac{1}{\left(N_{2}-N_{1}\right)}\sum_{k=N_{1}}^{N_{2}}{\left[x_{mes}\left(k\right)-x_{est}\left(k\right)\right]}^{2}}$$
where $x_{mes}$ and $x_{est}$ represent the measurement and the estimates of the signal x(t), and $N_{1}$, $N_{2}$ denote the bounds including the transient or the steady states. Figure 7 shows the results obtained.

It is checked here that the estimate of speed in the transient state provided by the method of Lee is the most precise. It is also noted that the COMBINED method provides the best results on the three signals in steady state, as indicated in [27]. The results obtained on this motor and this test cannot be generalized. An exhaustive study must be carried out in order to explain the performance of the Lee method in transient conditions, but this is outside the context of this article. In the following, we retain the Lee model for the implementation of the TOT applied to our motor.

## 3. Specific Angular Re-Sampling for Rotor Asymmetry Monitoring

### 3.1. Angular Re-Sampling with Mechanical Position

Once the speed of rotation has been estimated, it is possible to deduce the mechanical position from a simple integration. This position is then used to carry out the angular re-sampling of the signal chosen to carry out the diagnosis. Offline, this sampling can be done in different ways [30]. As mentioned before, the quantities re-sampled with mechanical position are independent of velocity. Consequently, a conventional FFT calculated on these quantities becomes a powerful tool again, as at constant speed. The spectrum is no longer represented as a function of frequency (*f* in Hz) but as a function of a homogeneous variable with rad${}^{-1}$. The unit currently used is the number of events per turn (*g* in ev${}^{t}$/rev), also called *orders*. The relations between the angular variables and the specifications of the FFT are recalled in Table 3.

In this representation, for data sampled from the mechanical position, the component at the rotational frequency will be localized to the order g=1, whatever the speed variations. Considering the data sampled at the sampling period $\theta_{s}$, the spectrum will be calculated between g=0 and $g_{max}=\frac{1}{2\theta_{s}}$. The corresponding resolution is $\Delta_{g}=\frac{1}{N\theta_{s}}$. Each spectral component, due to faults or load variations, will be located at an order whose value will be constant. The angular sampling period $\theta_{s}$ will be chosen according to the maximum order to monitor. In the following, we will analyze the motor in transient state, i.e., during the start-up phase. The FFT resolution defined by $\Delta_{g}=\frac{1}{N_{\theta }\theta_{s}}$ will depend on the duration of this transitional regime.

In steady state, the angular spectrum expressed as a function of *g* is obtained simply from the time data by dividing the frequency scale by the value of the rotation frequency $F_{r}$. In a variable regime, the angle to use to re-sample the data is:(16)$$\theta_{r}\left(t\right)=2\pi \int_{0}^{t}F_{r}\left(\tau \right)d\tau$$

### 3.2. Design of a Specific Angle for Fault Monitoring

In the previous paragraph, we recalled the principle of angular sampling from the mechanical position. This principle can be generalized to any angle. For example, in the event of a rotor fault (bar breakage) and at fixed speed, the frequency components in the stator current will be located at [31,32]:(17)$$f=F_{e}\left(1\pm 2ks\right)=F_{e}\pm 2ksF_{e},$$
where $F_{e}$ is the supply frequency, *s* is the machine slip, and *k* is a positive integer.

It is known that this type of fault will generate amplitude and frequency modulations of the rotor currents [33]. It is therefore customary to look for defects in the quantities IA(t) or IF(t). In the modulation quantities IA(t) and IF(t), the fault components, due to the frequency translation around 0, will be located at
(18)$$f=2ksF_{e}=2kpF_{sync}\left(\frac{F_{sync}-F_{r}}{F_{sync}}\right)=2kp\left(F_{sync}-F_{r}\right),$$
with *p* denoting the number of pairs of poles of the motor and $F_{sync}=\frac{F_{e}}{p}$ the synchronism frequency.

In steady state (constant speed), the spectrum expressed as a function of the order could be obtained from a time-sampled signal by dividing the frequency scale by the quantity $2p(F_{sync}-F_{r})$. In the transient state, it is necessary to re-sample the analysis signal with a specific angle equal to:(19)$$\theta_{spec}\left(t\right)=2\pi \times 2p\int_{0}^{t}F_{sync}\left(\tau \right)-F_{r}\left(\tau \right)d\tau ,$$

After re-sampling analysis signals IA(t) or IF(t) with the specific angle $\theta_{spec}$, the defect components will be located at g=1, g=2, g=3…, whatever the motor speed (constant or variable speed). Due to the use of an observer, the quantities $F_{sync}$ and $F_{r}$ are accessible and the angle $\theta_{spec}$ can be estimated in real time. It is possible to generalize this approach to define the specific angles relative to the different possible defects (stator winding, bearings, torque oscillations, etc.).

## 4. Experimental Results

In these tests, we insert a faulty rotor into the motor. Two holes are made to simulate two bar breaks. The motor is started gradually over 5 s by varying the supply voltage amplitude with an auto-transformer. The observer provides the estimated rotation frequency (${\hat{F}}_{r}$) and the synchronism frequency ${\hat{F}}_{sync}=\frac{\hat{I}F}{p}$ calculated by (Equation 14). The specific angle (Equation 19) is then estimated in real time. This angle is then used to re-sample the signal $\hat{I}F\left(t\right)$, which is chosen as the analysis signal in this test. Re-sampling operation is performed by linear interpolation using the Matlab function “rpmordermap”. This function uses the following parameters:the signal to be analyzed: $\hat{I}F\left(t\right)$.the time sampling frequency: $F_{s}=10$ kHz.the motor speed expressed in rpm: we adapt this function to use it with our specific angle by entering the speed $N=60\left[{\hat{F}}_{sync}\left(t\right)-{\hat{F}}_{r}\left(t\right)\right]$. This choice corresponds to choosing a reference frame rotating at the frequency of the rotor currents.

Finally, the Matlab function “orderspectrum” is used to calculate the spectrum of the signal. Figure 8 shows the complete procedure used in the following tests. The same procedure is applied to analyze the signal IF(t) resulting from the measurements.

The first test concerns the analysis of signal IF(t) (calculated from measurements) and $\hat{IF}\left(t\right)$ (calculated from estimations) in the stationary zone (6 s < *t* < 16 s), as shown in Figure 9. On the steady-state spectrum (Figure 9b), we check the presence of the components at $2sF_{e}=4.35$ Hz and its multiples of a component at $F_{r}=23.9$ Hz, which is the frequency of rotation and which is probably due to motor misalignment. There is also a spectral component at $F_{e}=50$ Hz, which is the translation of the fundamental component of the network at 100 Hz after demodulation. Figure 9c shows the spectrum of the signal $IF(\theta_{spec})$ sampled with the specific angle (19). As expected, the components related to the rotor fault are now located at g=1,2,3,4. The component at g=5.49 ev${}^{t}$/rev corresponds to the rotation frequency $F_{r}$ divided by $2sF_{e}$. The component at g=11.48 ev${}^{t}$/rev corresponds to $F_{e}$ divided by $2sF_{e}$.

We now perform an analysis of the same signal IF(t) in the transient zone (1 s < *t* < 6 s). The result is shown in Figure 10. The angular sampling from the specific angle $\theta_{spec}$ makes it possible to highlight the orders relating to the motor rotor fault. One notes the presence of components between g=0 and g=1, g=3, and g=4, which did not appear in steady state. These components are related to the transitory regime, which generates additional spectral components.

Since the angular spectrum is now stationary (independent of the speed), it is possible to carry out a simple filtering to isolate the components sought. Time Synchronous Averaging (TSA) type filtering is applied [34]. The frequency response of the filter is of the form:(20)$$F\left(g\right)=\frac{1}{M}\frac{sin(\pi Mg)}{sin(\pi g)}$$

The function (Equation 20) is a sequence of cardinal sine centered on g=1,2,3…. The width of the lobes is regulated by the coefficient *M*. The filtered signal is obtained by a simple multiplication of the original spectrum by the function F(g). The result is given in Figure 10b for M=10. The components linked to the fault are correctly isolated by the TSA filtering.

## 5. Discussion and Conclusions

In this article, we presented a methodology allowing us to carry out an analysis of the electrical currents of an asynchronous machine in transitory state starting from a reduced amount of information. The known methods for Tacholess Order Tracking have been little used in the case of induction motors because they require an estimation of the rotational speed. Model-based speed estimation methods require the identification of dynamic motor parameters, which makes them difficult to use in an industrial environment. From this point of view, the strategy proposed in this article is a real contribution for the development of angular sampling techniques for squirrel-cage induction motors. The first important point is the possibility of establishing an observer from the nameplate motor without resorting to an identification procedure. We have shown that the COMBINED method does not give the best results in transient state. Although, for this study, the Lee method is the most precise, it is impossible to generalize this result to all motors. An additional study is in progress to justify this result and generalize the methodology. From an industrial point of view, the fact that the Lee method gives good results is interesting because it is the one that mobilizes the least information to establish the static model. Another advantage in the use of an observer is that it gives access to other quantities on which it is possible to search for faults—in particular, speed and flux. The second interesting point is the generalization of the angular re-sampling notion. Although it is generally applied from the mechanical position, we have shown that it is possible to define an angle specific to the desired defect. In this article, we have dealt with the case of a rotor fault, but we could extend the principle to all possible motor faults (short circuit to the stator, bearings, etc.) and also to the frequency components generated by the load and its coupling (gearbox, belt, etc.). This generalization is also the subject of current work. Finally, in this work, we used re-sampling by linear interpolation. It is planned to apply our methodology with more efficient techniques (Kalman, Vold–Kalman, and Time-Variant Discrete Fourier Transform).

## Figures and Tables

**Figure 1 sensors-22-03371-f001:**
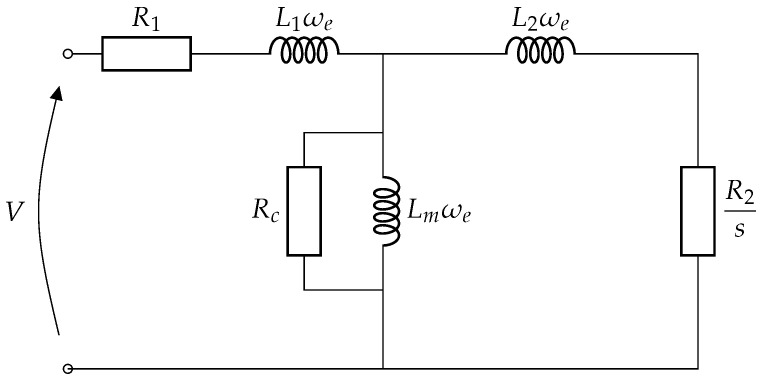
Squirrel-cage induction motor model in steady state.

**Figure 2 sensors-22-03371-f002:**
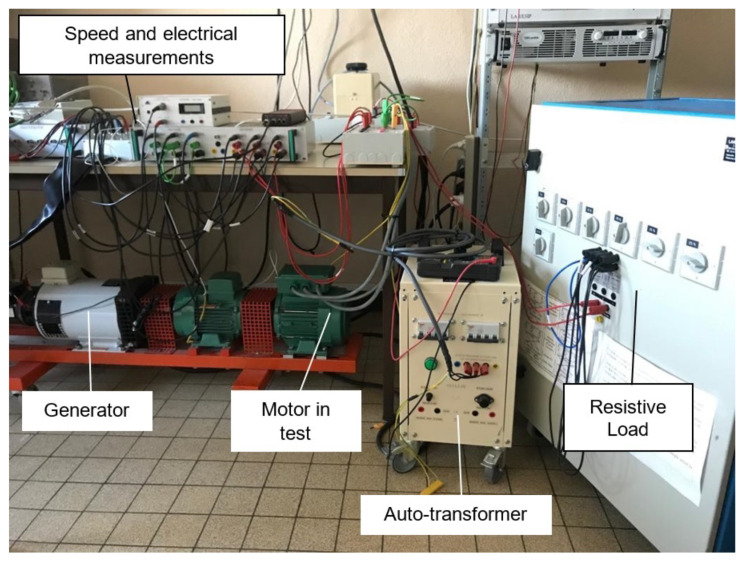
Experimental set-up.

**Figure 3 sensors-22-03371-f003:**
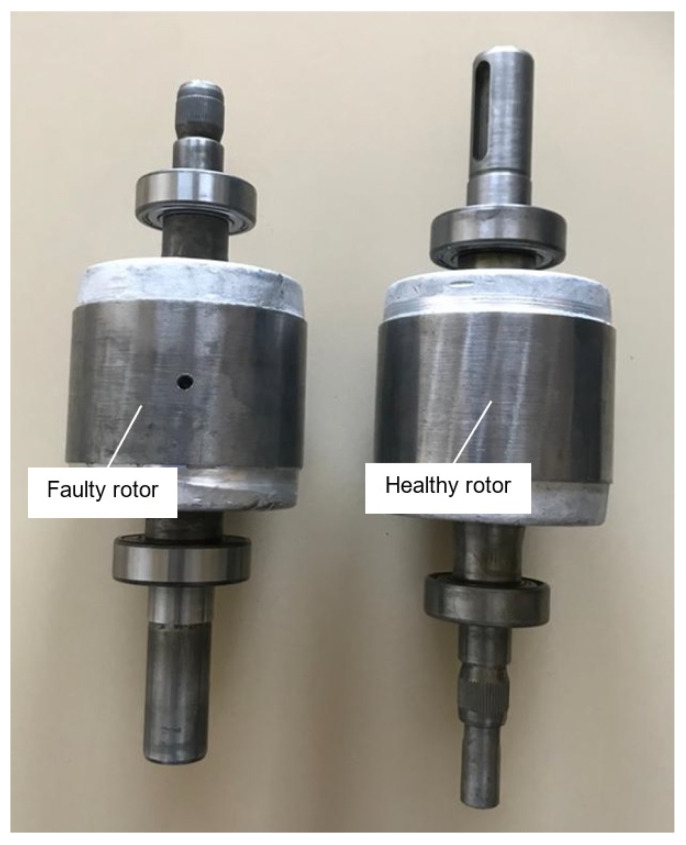
Healthy and faulty rotors.

**Figure 4 sensors-22-03371-f004:**
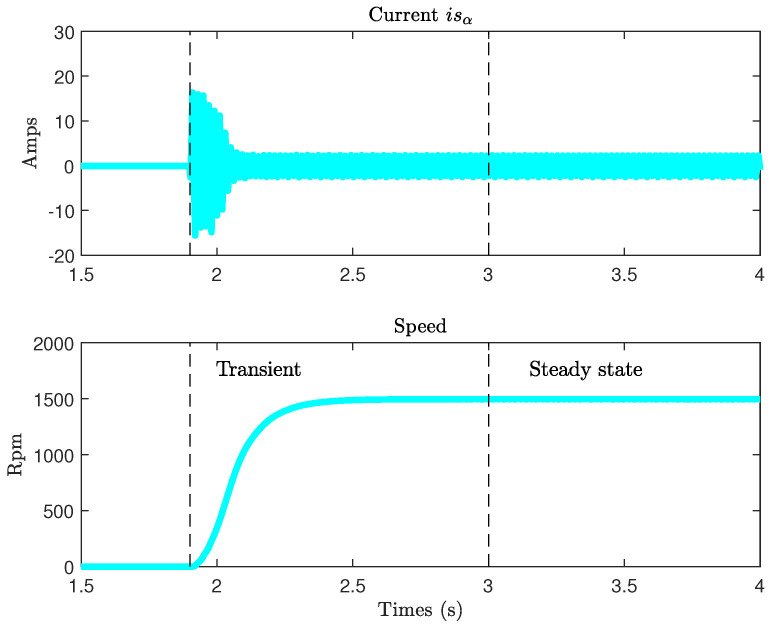
Direct Concordia current $i_{s\alpha (t)}$ and speed.

**Figure 5 sensors-22-03371-f005:**
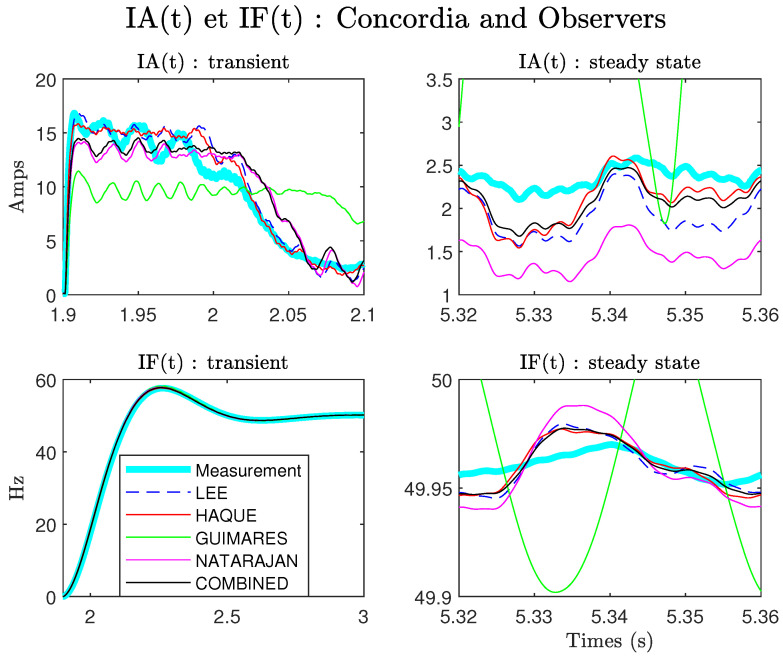
Comparison between models: IA(t) et IF(t), $K_{i}=$ 10,000, $K_{p}=0$, G=0.

**Figure 6 sensors-22-03371-f006:**
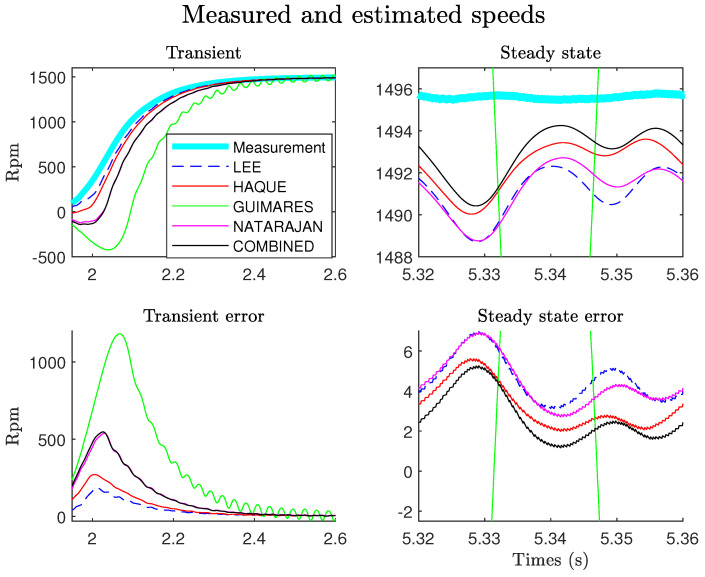
Comparison between measured and estimated speeds, $K_{i}=$ 10,000, $K_{p}=0$, G=0.

**Figure 7 sensors-22-03371-f007:**
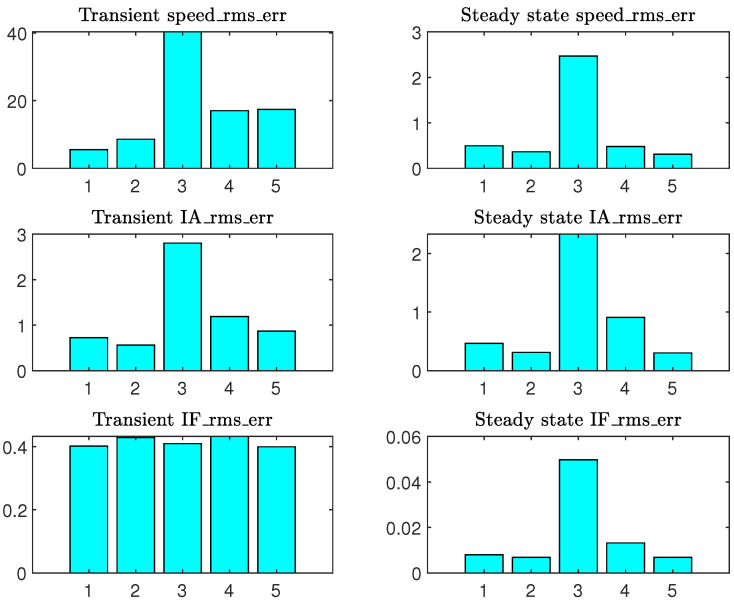
Criterion (15) calculated for speed, IA(t) and IF(t): (1) Lee, (2) Haque, (3) Guimarães, (4) Natarajan, (5) COMBINED.

**Figure 8 sensors-22-03371-f008:**
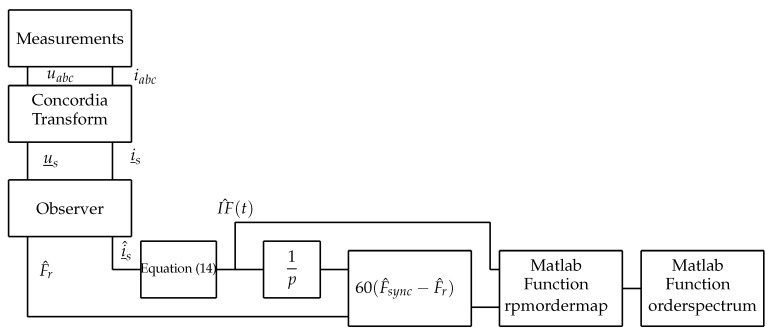
Complete TOT procedure.

**Figure 9 sensors-22-03371-f009:**
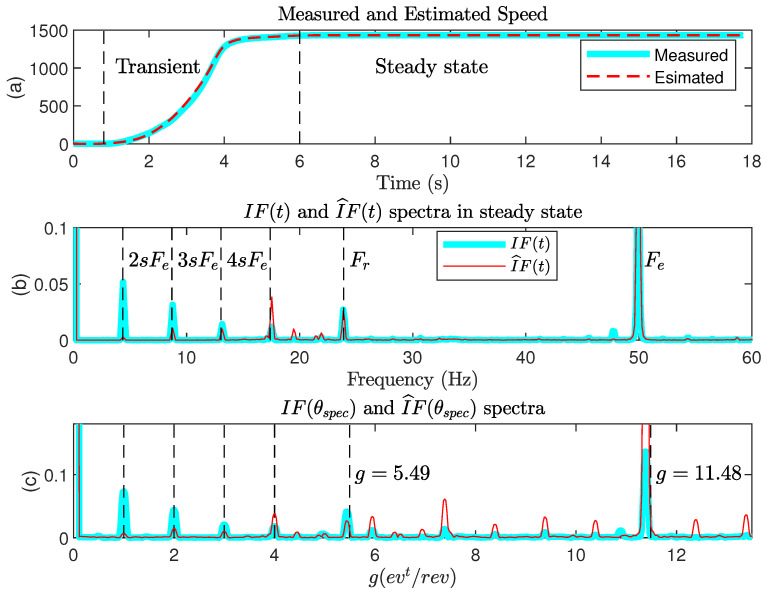
IF(t) and $\hat{I}F\left(t\right)$ analysis in steady state.

**Figure 10 sensors-22-03371-f010:**
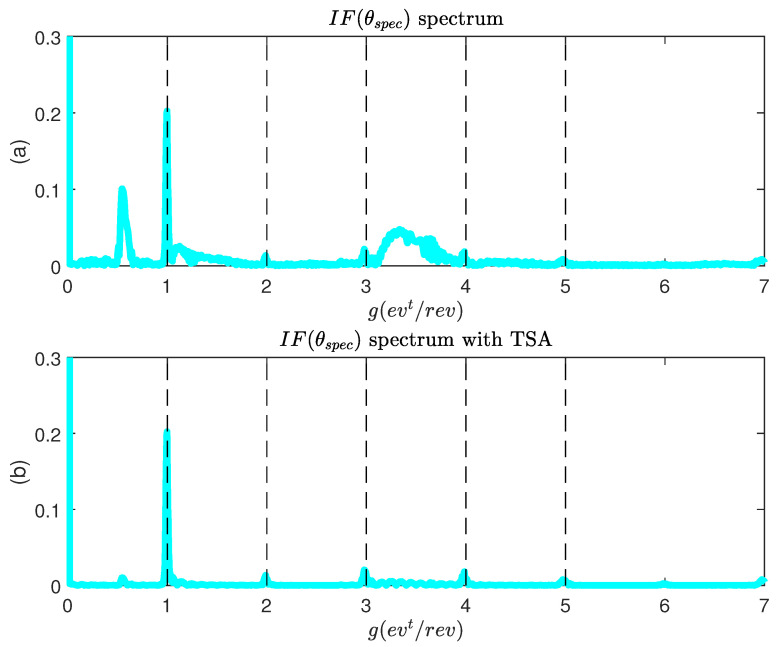
IF(t) analysis during start-up.

**Table 1 sensors-22-03371-t001:** Data needed for the different methods.

Model	NP	cosφ			η			Torque		Current	NEMA	
		**100**	**75**	**50**	**100**	**75**	**50**	$T_{d}$	$T_{m}$	$I_{d}$	**CODE**	**CLASS**
Natarajan-Misra [23]	*	*	*	*	*	*	*	*	*		*	
Lee [25]	*									*	*	*
Haque [24]	*	*		*	*		*		*		*	
Guimarães [26]	*	*	*	*	*	*	*	*	*	*		
Amaral [28]	*	*	*	*	*	*	*				*	

**Table 2 sensors-22-03371-t002:** Steady-state models.

Method	$R_{1}$	$X_{1}$	$R_{2}$	$X_{2}$	$R_{c}$	$X_{m}$
Lee	10.06	6.26	4.96	9.39	978.82	179.38
Haque	12.99	4.83	5.16	7.25	1619.1	153.76
Guimarães	13.18	3.23	4.05	31.88	1778.3	181.04
Natarajan	12.67	7.70	4.55	11.55	3985.43	232.96
Amaral	13.16	7.60	4.06	11.35	5289.16	160.96
COMBINED	13.18	7.60	4.55	11.35	1619.1	153.76

**Table 3 sensors-22-03371-t003:** Angular domain and FFT notations.

Angular Domain			
**Name**	**Variable**	**Expression**	**Unit**
$\theta_{s}$	Sampling period	-	rd
$G_{s}$	Sampling frequency	$\frac{2\pi }{\theta_{s}}$	Ev${}^{t}$/rev
$N_{\theta }$	Number of samples	-	-
$g_{max}$	FFT domain	$\frac{G_{s}}{2}$	Ev${}^{t}$/rev
$\Delta_{g}$	FFT resolution	$\frac{G_{s}}{N_{\theta }}$	Ev${}^{t}$/rev

## Data Availability

Samples of the compounds are available from the authors.

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
