# Peer review of "Transient Detection of Rotor Asymmetries in Squirrel-Cage Induction Motors Using a Model-Based Tacholess Order Tracking"

_sensors, 2022, doi:10.3390/s22093371_

Round 1

Reviewer 1 Report

Comments in Word file.

Author Response

Dear Colleague,

thank you for your remarks very constructives.  Please find below our answers in the attached file.

Kind regards

E.E

Reviewer 2 Report

The paper entitled "Transient detection of rotor asymetries in induction motors using a model-based Tacholess Order Tracking (TOT)" . The paper is well written however i have certain comments on the work. 

  1. The paper abstract should only have information about the work. Please remove all the introductory sentences in the abstract.
  2. This paper seems to be a comparative analysis of few techniques. I suggest that the flowcharts of all the methods be made part of the paper. 
  3. Please explain the experimental setup and add photos of setup along with the annotations. 
  4. In connection with comment 2, please clearly explain the contributions of this work. 

Author Response

(The authors gave the same response as above.)

Reviewer 3 Report

This paper proposes deals with the topic of monitoring of asynchronous machines by analyzing transient electrical signals. However, the rotor speed must be determined accurately. The authors propose determine the dynamic model of the induction machine from a reduced amount of formation, which is used for rotor fault monitoring purposes. The proposed method is validated in the laboratory using a test bench dedicated to the study of rotor bar defects.

I suggest a revision based on the following points in order to improve the quality of the paper:

1.- Introduction Section. The novelties and contributions of the paper must be further stressed in the end paragraphs of the Introduction sections, as well as the advantages and disadvantages (if any) of the proposed approach.

2.- Figure 1. The magnitudes of the rotor must be transferred to the stator.

3.- Methods summarized in Table 1 require a proper reference. In addition, the paper is based on these methods, but no information or a summary of their application is not provided in the text.

4.- Results presented in Table 2 present a high dispersion degree. Further comments are required as an explanation how it affects the accuracy of the method proposed in this work.

5.- Please include units in the vertical axis of Figs. 3 and 4.

6.- Fig. 6. Please translate Fréquence into Frequency.

7.- Fig. 6. It is suggested to include the estimated spectra.

I hope this revision can help the authors to improve the quality and readability of the paper.

Author Response

(The authors gave the same response as above.)

Reviewer 4 Report

The methodology presented in the paper is interesting, but I think that some improvements are needed:

1)  Correct many spelling mistakes (e.g., "asymetries" in the title, "mtor" at line 24, "??" at line 50, "=" at line 228, ...) and some French words (Table 1, line 145, Table 2, title of Section 3, figure 6).

2) The acronym TOT should not be used in the title. Moreover, the definition of TOT in the Abstract is not correct.

3) Consider to use "angular frequency" instead of "pulsation" throughout the paper.

4) Add the definition of the slip s in the list below figure 1.

5) Provide proper references for the methods listed in Table 1.

6) Section 4 with experimental results should be expanded. In fact, in Section 5 it is stated that the principle could be extended to all possible motor faults. At least one further case should be investigated (e.g., short circuit to the stator) to improve the quality and the thoroughness of the paper.

Author Response

(The authors gave the same response as above.)

Round 2

Reviewer 3 Report

Please include units in the vertical axis of Figs. 4

Some equations must be reformated

Author Response

Dear colleague,

Thank you for your comments. Corrections have been made to the text. All notations under the equations have been aligned left.

Best regards

E. ETIEN

Reviewer 4 Report

The Authors should check and correct misalignment of text and equations throughout the paper.

Author Response

(The authors gave the same response as above.)
